

# Soil microbial community and physicochemical properties together drive soil organic carbon in *Cunninghamia lanceolata* plantations of different stand ages

Ye Yuan, Juan Li and Liang Yao

School of Ecology and Environment, Anhui Normal University, Wuhu, China

Corresponding author
Liang Yao, dylanyao@126.com

## ABSTRACT

Carbon sequestration in forest soil is critical for reducing atmospheric greenhouse gas concentrations and slowing down global warming. However, little is known about the difference in soil organic carbon (SOC) among different stand ages and the relative importance of biotic and abiotic variations such as soil microbial community and soil physicochemical properties in the regulation of SOC in forests. In the present study, we measured the SOC of the topsoil (0-10 cm) in Chinese subtropical *Cunninghamia lanceolata* plantations of three different stand ages (young plantation of 6 years, middle-aged plantation of 12 years, and mature plantation of 25 years). We further measured microbial community composition by phospholipid fatty acid (PLFA) analysis and soil organic carbon physical fractions by wet sieving and density floating as well as other physicochemical properties. The effects of the main impact factors on SOC were investigated. The results showed that: the middle-aged plantation had significantly higher SOC (10.63 g kg$^{-1}$) than the young plantation (5.33 g kg$^{-1}$), and that of the mature plantation (7.83 g kg$^{-1}$) was in between. Besides, the soil total PLFAs and all the functional groups (*i.e.*, bacteria, fungi, actinomycetes, Gram-positive bacteria, and Gram-negative bacteria) of PLFAs were significantly higher in the middle-aged plantation than in the young plantation and the mature plantation. Soil physicochemical properties, including physical fractions, differed among plantations of the three stand ages. Notably, the proportion of organic carbon protected within microaggregates was significantly higher in the middle-aged plantation (40.4%) than those in the young plantation (29.2%) and the mature plantation (27.8%), indicating that the middle-aged *Cunninghamia lanceolata* plantation had stronger soil organic carbon stability. Both soil microbial community and physicochemical properties exerted dominant effects on SOC and jointly explained 82.7% of the variance of SOC among different stand ages. Among them, total and all the functional groups of PLFAs, nitrate nitrogen, total nitrogen, and organic carbon protected within microaggregates had a significant positive correlation with SOC. These results highlight the important role of soil biotic and abiotic factors in shaping the contents of SOC in forests of different stand ages. This study provides a theoretical basis for forestry management and forest carbon cycling models.

Physicochemical properties, Soil organic matter fractions

# INTRODUCTION

Global warming caused by greenhouse gas enrichment is changing the structure, functions and some key ecological processes of the terrestrial ecosystem (*Jones et al., 2009*), thus has become the focus of global change research (*Peters et al., 2013*). The global forest, with a coverage area of $4.0 \times 10^9$ hectares, is the main carbon pool of terrestrial ecosystems (*Pan et al., 2011*), and plays an important role in regulating the global carbon cycle and climate change (*He, 2012*). Afforestation and plantation management are the main ways to increase forest cover, thereby fixing more $CO_2$ from the atmosphere and slowing down global warming. Plantation management, such as harvesting and forest fires, could create plantations of different stand ages. Stand age and corresponding biomass were found to be the most important factors in determining the net primary production (NPP) of forest plantations (*Michaletz et al., 2014*). What's more, soil organic carbon (SOC) content (*Hou et al., 2019*) and its decomposition (*Wu et al., 2020b*) were also found to vary with stand age. How does soil organic carbon in subtropical plantations varies with stand age and what are the main drivers need more research.

It is worth noting that SOC could be partly protected from decomposition by being encapsulated in aggregates or adsorbed by minerals (*Krull, Baldock & Skjemstad, 2003*). Therefore, according to the stability and turnover time, SOC could be divided into different components, including mineral-associated organic carbon (SC), organic carbon protected within microaggregates (SA), and particulate organic matter (POM) (*Mitchell et al., 2018*). However, most studies about SOC in forests focused on its total content (*Guan et al., 2019*; *Thom et al., 2019*) and there is still a lack of studies about SOC fractions. We assume that the ratio of different components of SOC as well as other soil physicochemical properties will affect SOC storage. Soil microbes are an important part of the soil ecosystem, affecting soil fertility, nutrients, and carbon cycling processes (*Soong et al., 2020*). Plantations of different stand ages have diverse NPP (*Anderson-Teixeira et al., 2016*), litter, and root exudates (*Wu et al., 2020a*), which could directly lead to microbial differences. Previous studies have found that microbial community composition plays an important role in the formation and persistence of soil organic matter (*Condron et al., 2010*). Therefore, we assume that microbial community is one of the most important drivers of SOC among plantations of different stand ages.

There is a large area of artificial forests in China. *Cunninghamia lanceolata* is an excellent fast-growing native species, which is not susceptible to pests and diseases and is widely used in commercial wood. *Cunninghamia lanceolata* has the largest planting area ($8.5 \times 106$ hm$^2$) among China's artificial forests and it accounts for 21.4% of the total forest area in China based on the seventh national forest inventory statistics. In this study, we compared SOC in *Cunninghamia lanceolata* plantations of three different stand ages (young plantation of 6 years, middle-aged plantation of 12 years, and mature plantation of

25 years) in subtropical China. To explore the key determinant of SOC, we further detected microbial community composition, SOC physical fractions, and other physicochemical properties. We then conducted variation partitioning analysis (VPA) to quantify the relative contribution of microbial community and physicochemical properties to the variations of SOC among different stand ages. The results will reveal patterns and drivers of SOC in different developmental stages of *Cunninghamia lanceolata* plantations and provide a theoretical basis for the scientific management of the artificial forest.

## MATERIALS & METHODS

### Study site and soil sample collection

Our study was conducted at Gaojingmiao Forest Farm (31°00′59″N, 119°12′08″E) in Langxi County, Xuancheng City, Anhui Province, China. Gaojingmiao Forest Farm is a state-owned forest farm, covering an area of 10.37 km², with large areas of *Pinus massoniana*, *Pinus elliottii*, and *Cunninghamia lanceolata*. This forest farm is located in the southeast of Anhui Province and has a humid subtropical monsoon climate with four distinct seasons and sufficient sunlight. The mean annual temperature in this area is 15.9 °C, and the coldest and warmest months are January (with a mean temperature of 2.7 °C) and July (with a mean temperature of 28.1 °C), respectively. The region receives an average total annual precipitation of 1,294.4 mm mostly occurring in summer. The area has 1784 h of sunshine and 228 days of Frost-free period per year. The soil is acidic yellow-red soil.

In December 2018, *Cunninghamia lanceolata* plantations of three stand ages (young plantation of 6 years, middle-aged plantation of 12 years, and mature plantation of 25 years) with basically the same site conditions (30–50 m above sea level, slope less than 30° ) were selected as the sample blocks. Three plots (10 m × 10 m) were randomly set in each of the stand ages of *Cunninghamia lanceolata* plantation. Five soil cores were randomly collected from the 0–10 cm soil layer in each plot using a corer six cm in diameter. These five soil cores were then combined and mixed thoroughly as one sample for each plot. The individual soil samples of each plot were collected in plastic bags and placed in a cooler in the field, then transported to the laboratory for further analysis.

After visible plant residues and stones were removed, each soil sample was divided into two subsamples. One subsample was used to determine pH, soil organic carbon (SOC), and total nitrogen (TN) content after air-dried; one subsample was stored at −20 °C and used to determine microbial community composition, ammonium nitrogen ($NH_4^+$-N), nitrate nitrogen ($NO_3^-$-N), and dissolved organic carbon (DOC) (*Yuan et al., 2020*). Cylinders were used to obtain four undisturbed soil cores from the 0–10 cm profile of each plot. One of these soil cores was used to determine soil bulk density (BD) and soil water content (SWC), while the other three of them were used to determine SOC physical fractions. The three soil cores used to determine SOC fractions were gently broken apart along the natural breakpoints and passed through a two mm sieve to remove visible organic debris and stones, and then air-dried for further analysis (*Mitchell et al., 2018*).

## Soil physicochemical properties analysis

Both SOC and TN were measured using a C/N analyzer (Elementar, Vario Max CN, Germany) with a combustion temperature of 900 °C. Soil samples collected with cylinders were oven-dried to calculate BD and SWC according to the volume of the cylinder and the soil weight before and after drying. Soil pH was measured at a soil: water ratio of 1:2.5 using a pH meter (Mettler Toledo, Greifensee, Switzerland). Soil $NH_4^+$-N and $NO_3^-$-N were extracted from 20 g of fresh soil with 1 mol $L^{-1}$ KCl (soil: extract, 1:5) and analyzed using a flow-injection autoanalyzer ((CFA)-AA3, SEAL, Germany). Soil DOC was extracted using deionized water (soil: extract = 1:5) and analyzed using a TOC analyzer (Vario TOC cube; Elementar, Hanau, Germany).

Wet sieving and density flotation methods were used to detect SOC fractions (*Mitchell et al., 2018*). First, soil samples were dispersed by low-energy sonication. Briefly, add 30 g of air-dried soil into 150 mL water and then put them into an ultrasonic treatment (KQ5200DE; Kunshan Ultrasound Instrument Co., Ltd., China) with the output energy of 22 J $ml^{-1}$. The macroaggregates were disrupted in this step and only the more stable microaggregates were left. Secondly, the dispersed suspension was then wet sieved over a 53 μm aperture sieve several times until the rinsing water was clear. Soil samples were separated into two parts in this step: the fraction >53 μm which was left on the sieve and the fraction <53 μm which passed through the sieve and remained in suspension. The fraction >53 μm was rinsed repeatedly with deionized water, dried in the oven at 40 °C, and weighed. Thirdly, density flotation was conducted to separate the fraction >53 μm from the second step into organic carbon protected within microaggregates (SA) and particulate organic matter (POM). The fraction >53 μm from the second step was stirred with sodium polytungstate (SPT) at a density of 1.8 g $cm^{-3}$ and then centrifuged at 1,000 g for 15 min. The light part was POM, which could be decanted and then washed with deionized water to remove all SPT, dried at 40 °C, and weighed. The heavy part was also dried at 40 °C and weighed as SA. At last, the fraction <53 μm from the second step was filtered through a 0.45 μm aperture nylon mesh, and the material >0.45 μm was dried at 40 °C and weighed as mineral-associated organic carbon (SC).

## Phospholipid fatty acid (PLFA) analysis

The PLFA contents of the samples were analyzed using the method described by *Bååth & Anderson (2003)*. Briefly, lipids were extracted from freeze-dried soil (8 g) in a single-phase mixture of chloroform:methanol:phosphate buffer (1:2:0.5). After extraction, the lipids were separated into neutral lipids, glycolipids, and polar lipids (phospholipids) on a silicic acid column. The samples were analyzed using a Thermo ISQ gas chromatography-mass spectroscopy system (TRACE GC Ultra ISQ). The concentrations of the individual compounds were obtained by comparing the peaks with an internal standard (nonadecanoic acid methyl ester 19:0). The fatty-acid signatures 15:0, i15:0, a15:0, i16:0, 16:1 ω7c, 17:0, i17:0, cy17:0 and cy19:0 were used as bacterial biomarkers (B). The fatty acids 18:2 ω6 and 18:1 ω9c were used as fungal indicators (F) and 10Me16:0 and 10Me18:0 were used as indicators for the actinomycetes (ACT). The fatty acids i15:0, a15:0, i16:0, and i17:0 were used to represent Gram-positive bacteria (G$^+$), while 16:1 ω7c, cy17:0, and cy19:0 were

used to represent Gram-negative bacteria ($G^-$). The total PLFAs (TPLFA) were the sum of bacterial, fungal, and actinomyces PLFA biomarkers.

## Statistical analysis

A one-way ANOVA was performed to identify significant differences in SOC, soil physicochemical properties, and PLFAs among the three stand ages. A principal components analysis (PCA) was used to analyze soil microbial community structure. The Pearson correlation method was used to analyze the correlation between SOC and some important physicochemical properties and PLFAs. One-way ANOVA and Pearson correlation analysis was performed using SPSS 19.0 (SPSS Inc., Chicago, IL, USA). PCA was conducted with CANOCO for Windows 4.5. Other figures were generated using the Origin 9.5 package (Origin Lab Corporation, Northampton, MA, USA). $P < 0.05$ was considered to be statistically significant. Mean ±standard deviation was shown in tables and figures.

Variation partitioning analysis (VPA) was conducted to quantify the relative importance of soil microbial community and soil physicochemical properties to the variations of SOC among *Cunninghamia lanceolata* plantations of different stand ages. To avoid collinearity, variables with a large variance inflation factor (VIF) were removed in order until the VIF of the retained variables does not exceed 10. Next, principal component analysis was performed on microbial variables and physicochemical variables respectively (the first two principal components were retained). Then VPA was employed to quantify their explanation for the variations of SOC among different stand ages. VPA was conducted in R 4.0.2 (*R Core Team, 2020*) and the vegan package (*Oksanen et al., 2020*).

# RESULTS

## Soil organic carbon content

Soil organic carbon content exhibited large variability among different stand ages (Fig. 1). The middle-aged plantation had the highest SOC (10.63 g kg$^{-1}$) followed by the mature plantation (7.83 g kg$^{-1}$), and the young plantation had the lowest SOC (5.33 g kg$^{-1}$). Soil organic carbon content of the middle-aged plantation was 99.4% and 35.8% higher than those of the young and the mature plantation, respectively. The difference in SOC between the middle-aged and the young plantations was significant.

## Soil physicochemical properties

Soil physicochemical properties of the *Cunninghamia lanceolata* plantations varied among different stand ages (Table 1). The middle-aged plantation had the highest pH, SWC and TN. The mature plantation had a significantly higher $NH_4^+$-N content than the other stand ages. Soil BD, C/N, $NO_3^-$-N, and DOC showed no significant difference among the three stand ages.

Soil organic carbon fractions were different among *Cunninghamia lanceolata* plantations of different stand ages (Fig. 2). The proportion of SA was significantly higher in the soil of the middle-aged plantation (40.4%) than those of the young (29.2%) and the mature (27.8%) plantation. The other two fractions, SC and POM, showed no significant difference among the three stand ages.

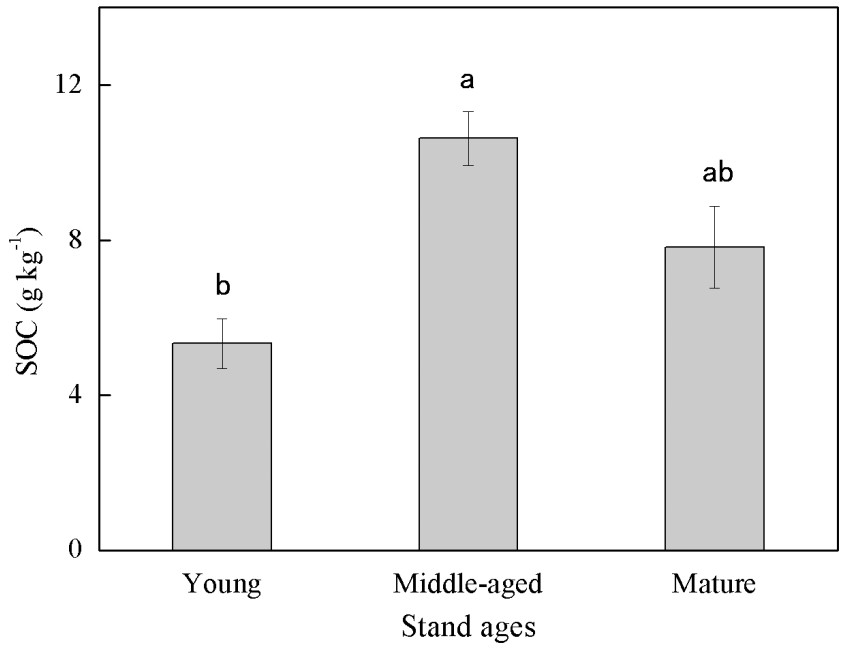

**Figure 1** **Soil organic carbon (SOC) content in Cunninghamia lanceolata plantations of three different stand ages.** Different lowercase letters indicated a significant difference among plantations of different stand ages at 0.05 level.

**Table 1** **Soil physicochemical properties in *Cunninghamia lanceolata* plantations of three different stand ages.**

| Soil physicochemical properties | Young plantation | Middle-aged plantation | Mature plantation |
|---|---|---|---|
| pH | $4.88 \pm 0.03b$ | $5.10 \pm 0.08a$ | $4.67 \pm 0.06b$ |
| SWC(%) | $20.83 \pm 0.46ab$ | $22.50 \pm 0.77a$ | $19.74 \pm 0.41b$ |
| BD(g cm$^{-3}$) | $1.07 \pm 0.14a$ | $1.30 \pm 0.03a$ | $1.33 \pm 0.04a$ |
| TN(g kg$^{-1}$) | $0.60 \pm 0.06b$ | $1.07 \pm 0.03a$ | $0.67 \pm 0.07b$ |
| C/N | $9.35 \pm 1.50a$ | $9.96 \pm 0.48a$ | $11.34 \pm 0.35a$ |
| NH$_4^+$-N(mg kg$^{-1}$) | $12.37 \pm 0.98b$ | $15.27 \pm 2.10b$ | $22.13 \pm 1.25a$ |
| NO$_3^-$-N(mg kg$^{-1}$) | $0.68 \pm 0.13a$ | $1.55 \pm 0.34a$ | $1.40 \pm 0.28a$ |
| DOC(mg kg$^{-1}$) | $17.67 \pm 6.13a$ | $14.74 \pm 2.92a$ | $6.44 \pm 1.15a$ |

Notes.

The data shown in the table is the mean ± standard error ($n = 3$). Different lower case letters indicated a significant difference among plantations of different stand ages at 0.05 level.

SWC, Soil water content; BD, Bulk density; TN, Total nitrogen; C/N, the ratio of organic carbon to total nitrogen; NH$_4^+$-N, Ammonium nitrogen; NO$_3^-$-N, Nitrate nitrogen; DOC, Dissolved organic carbon.

## Soil microbial community

Soil PLFAs showed significant differences among different stand ages (Fig. 3). The middle-aged plantation had significantly higher soil total PLFAs, bacterial PLFAs, fungal PLFAs, actinobacterial PLFAs, Gram-positive and Gram-negative bacterial PLFAs than the young plantation and the mature plantation (Fig. 3A). Soil total PLFAs of the middle-aged plantation were 30.23 nmol g$^{-1}$, while those of the young plantation and the mature

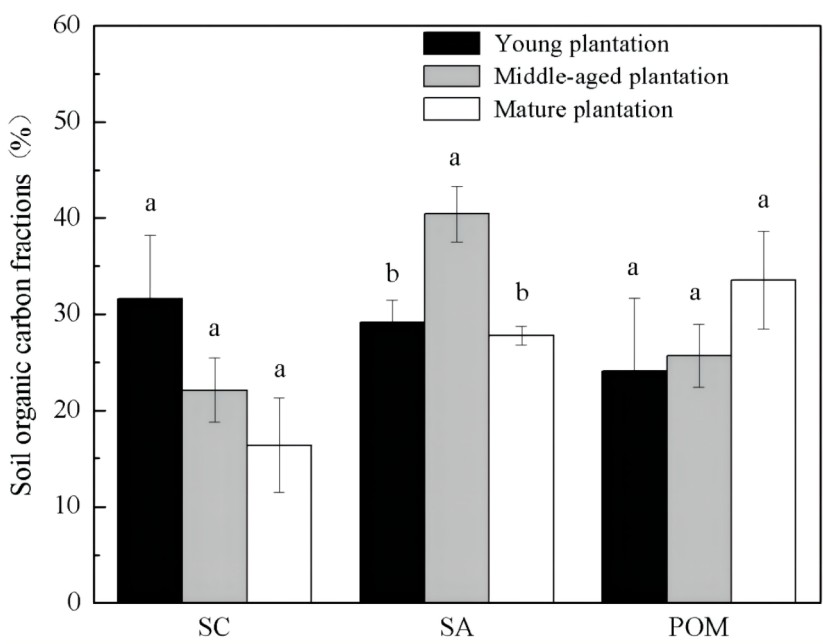

**Figure 2 Soil organic carbon physical fractions in *Cunninghamia lanceolata* plantations of three different stand ages.** SC, Mineral-associated organic carbon; SA, Organic carbon protected within microaggregates; POM, Particulate organic matter.

plantation were 72.5% and 60.3% lower than the middle-aged plantation. The ratio of F/B and the ratio of $G^+/G^-$ showed no significant difference among stand ages (Fig. 3B). Principal components analysis (PCA) also confirmed that soil microbial community composition was different among the three stand ages and 94% of the variance was explained by the first component of the PCA (Fig. 4). The soil PLFAs of the middle-aged plantation were separated from those of the other two stand ages along with the first principal component.

## Associations of SOC with the microbial community and physicochemical properties

The results of VPA showed that the soil microbial community and physicochemical properties explained 82.7% of the variance in SOC of different stand ages (Fig. 5). Pearson correlation analysis also showed that some of the microbial and physicochemical factors were significantly correlated with SOC (Table 2). Among the microbial factors, TPLFA, B, F, ACT, $G^+$, and $G^-$ were positively correlated with SOC while F/B was negatively correlated with SOC. Among the physicochemical properties, TN, $NO_3^-$-N, and SA were positively correlated with SOC.

## DISCUSSION

The SOC concentrations in this study (ranged from 5.33 to 10.63 g kg$^{-1}$, Fig. 1) were lower than those measured in some other subtropical *Cunninghamia lanceolata* plantations, which ranged from 18 to 25 g kg$^{-1}$ (*Chen et al., 2013*; *Song et al., 2017*). The possible

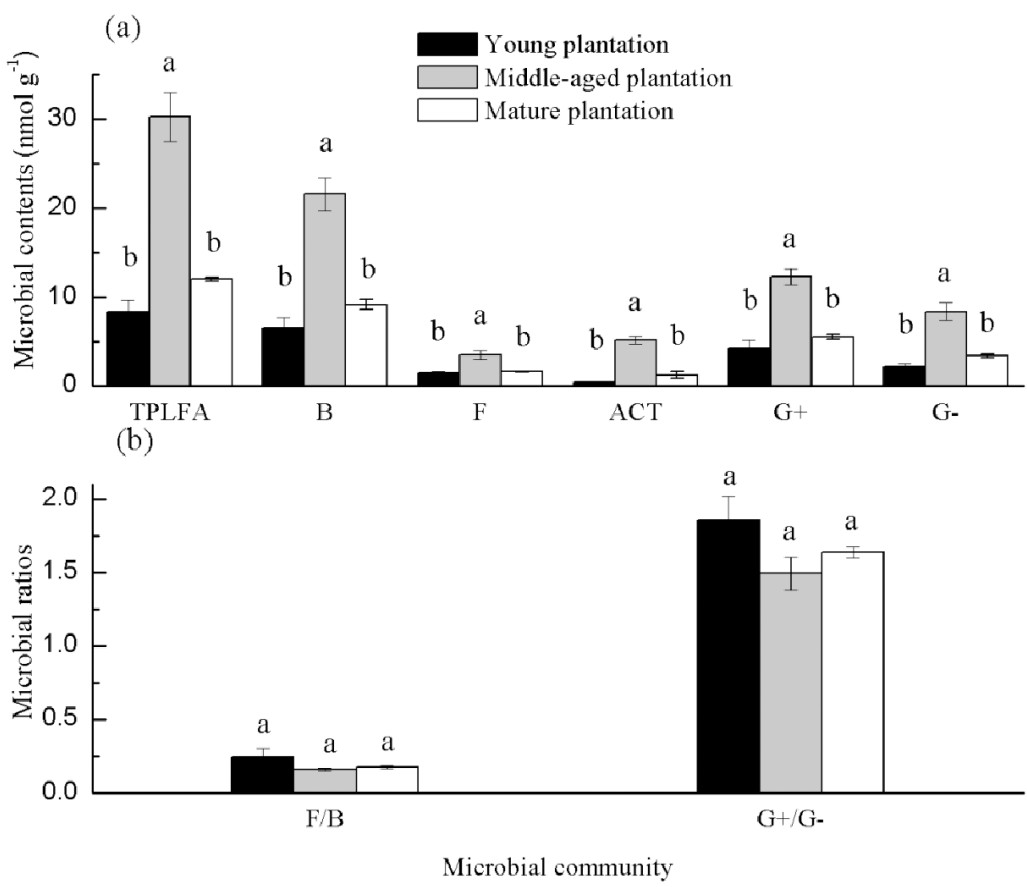

**Figure 3** **(A–B) Soil microbial community structure in *Cunninghamia lanceolata* plantations of three different stand ages.** TPLFA, Total phospholipid fatty acids; B, Bacteria; F, Fungi; ACT, Actinobacteria; $G^+$, Gram-positive bacteria; $G^-$, Gram-negative bacteria; F/B, Ratio of fungi to bacteria; $G^+/G^-$, Ratio of gram-positive bacteria to gram-negative bacteria.

reason for the lower SOC in our study is that the plantations here were converted from farmland decades ago, and the SOC of farmlands is generally lower than that of forest plantations (*Xie et al., 2007*). This implies that the forest plantations in this study area still have large carbon sequestration potential with proper management.

The middle-aged *Cunninghamia lanceolata* plantation had the highest SOC (Fig. 1) and the highest fraction of SA (Fig. 2). Compared with the POM fraction, the SOC occluded within microaggregates is more stable because microaggregates excluded microbes and enzymes from pores (*Yu et al., 2012*), indicating that the middle-aged plantation has more stable SOC. Forest SOC mainly comes from above-ground litter, fine root decomposition, and root exudates (*Berhongaray et al., 2019*). Young plantations, which are in the fast-growing stage, produce little litter. With the growth of the plantation age, the above-ground litterfall increased (*Zheng, Chen & Yan, 2019*). What's more, the increasing canopy closure will also accelerate the withering of understory vegetation (*Verburg, Johnson & Harrison, 2001*), and further increase the amount of surface litter. Most of the previous studies found

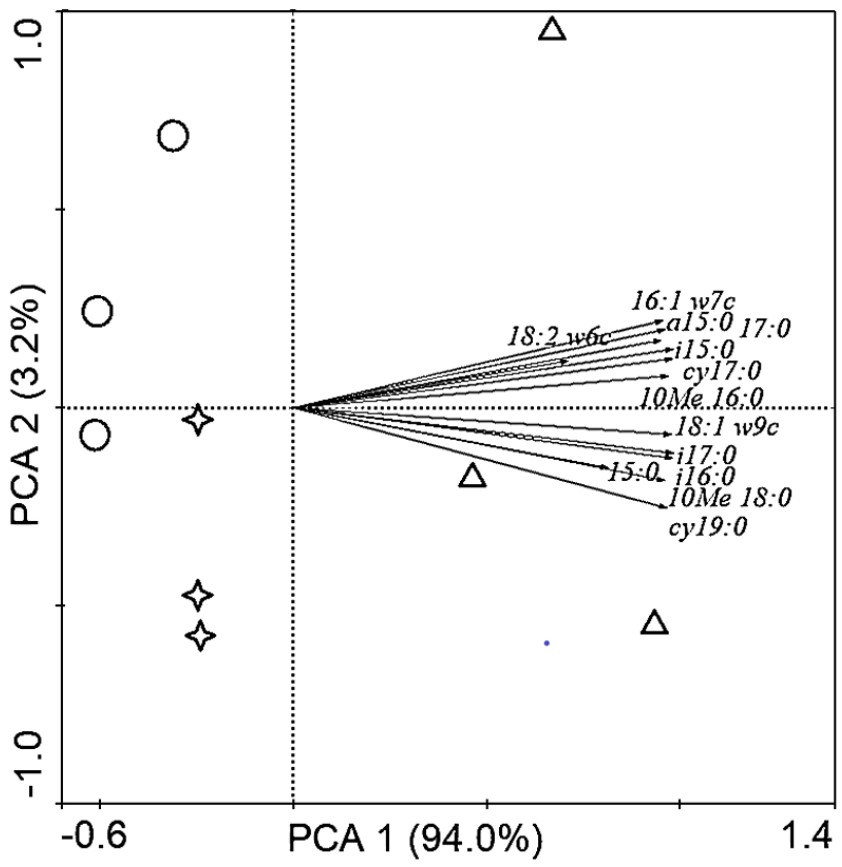

**Figure 4** **Principal component analysis (PCA) of soil microbial community structure in *Cunning-hamia lanceolata* plantations of different ages.** The red circles in the picture represent the young plantation, the blue triangles represent the middle-aged plantation, and the green stars represent the mature plantation.

that the mature forests had the highest SOC due to higher levels of litter accumulation and lower soil respiration (*Nath et al., 2022*; *Wang et al., 2019*). However, our study found that the SOC of mature plantations was lower than that of middle-aged plantations. This may be due to inactive root activities in the mature plantations. The decomposition process of litter takes a longer time, while the carbon input from root exudates to soil is more rapid. A previous study conducted in subtropical China showed that middle-aged *Cunninghamia lanceolata* plantations had higher underground carbon allocations than young and mature plantations (*Chen et al., 2008*). Because mature forests grow slowly and require fewer nutrients, they do not need as many roots for nutrients. The higher underground carbon allocations in middle-aged forests provided more organic carbon to the soil in the form of exudates and dead roots. While delivering large amounts of carbon directly to the soil, root exudate contains macromolecular viscose and promotes the formation of microaggregates (*Traoré et al., 2000*).

Soil total PLFAs and all microbial groups were highest in the middle-aged plantation (Fig. 3A). The possible reason is that the middle-aged plantations, had higher litter accumulation

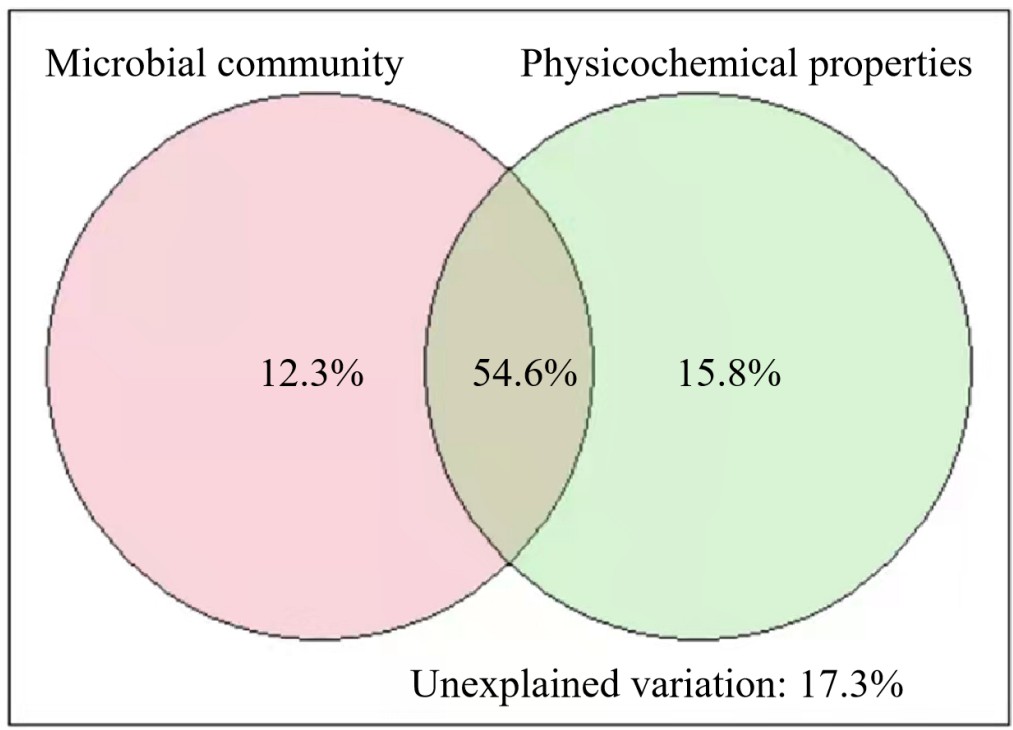

**Figure 5** Results of variation partitioning analyses illustrating the relative contribution of soil microbial community and soil physicochemical properties to soil organic carbon.

**Table 2** Pearson correlations between SOC and soil microbial/physicochemical properties.

| Microbial community | | | Physicochemical properties | | | | | |
|---|---|---|---|---|---|---|---|---|
| | *r*-value | *P*-value | | *r*-value | *P*-value | | *r*-value | *P*-value |
| TPLFA | **0.87** | **<0.001** | pH | 0.48 | 0.11 | SC | −0.22 | 0.50 |
| B | **0.87** | **<0.001** | SWC | 0.49 | 0.10 | **SA** | **0.62** | **0.03** |
| F | **0.72** | **0.008** | BD | 0.44 | 0.15 | POM | −0.22 | 0.50 |
| ACT | **0.89** | **<0.001** | **TN** | **0.90** | **<0.001** | | | |
| G$^+$ | **0.89** | **<0.001** | C/N | 0.32 | 0.31 | | | |
| G$^-$ | **0.85** | **<0.001** | NH$_4^+$-N | 0.34 | 0.29 | | | |
| F/B | **−0.68** | **0.02** | NO$_3^-$-N | **0.63** | **0.03** | | | |
| G$^+$/G$^-$ | −0.34 | 0.28 | DOC | 0.12 | 0.72 | | | |

**Notes.**

TPLFA, Total phospholipid fatty acids; B, Bacteria; F, Fungi; ACT, Actinobacteria; G$^+$, Gram-positive bacteria; G$^-$, Gram-negative bacteria; F/B, Ratio of fungi to bacteria; G$^+$/G$^-$, Ratio of gram-positive bacteria to gram-negative bacteria; SWC, Soil water content; BD, Bulk density; TN, Total nitrogen; C/N, the ration of organic carbon to total nitrogen; NH$_4^+$-N, Ammonium nitrogen; NO$_3^-$-N, Nitrate nitrogen; DOC, Dissolved organic carbon; SC, Mineral-associated organic carbon; SA, Organic carbon protected within microaggregates; POM, Particulate organic matter.

Bold numbers indicate significant correlations between SOC and soil properties.

than the young plantations and more root allocation than the mature plantations and this provided more carbon sources for the growth and reproduction of microbes. Higher microbial biomass generally corresponds to better soil structure and nutrient status (*Kang*

*et al., 2021*). The ratio of F/B could reflect ecosystem stability and soil nutritional status (*Liu et al., 2019*). The ratios of F/B were all smaller than 1 (Fig. 3B), indicating the absolute dominance of bacteria. And in the present study, there was no significant difference in the ratio of F/B among plantations of different stand ages. The ratios of $G^+/G^-$ were all bigger than 1 (Fig. 3B). This may be due to the different water requirements of $G^+$ and $G^-$ bacteria, the subtropical humid soil environment is more suitable for the growth and reproduction of $G^+$ bacteria (*Zhou et al., 2017*).

Our results illustrated the roles of soil microbial community and physicochemical properties in regulating SOC across different stand ages (Fig. 5). We found that all microbial groups (PLFAs) were positively correlated with SOC (Table 2). While SOC provides food sources for microorganisms, soil microorganisms and their debris are important components of SOC (*Guo et al., 2021*). The higher root exudates of middle-aged plantations provided more carbon sources for the growth and reproduction of microorganisms and produced more microbial necromass C to SOC. Besides, soil physicochemical properties, especially TN, $NO_3^-$-N, and SA, were also found to regulate SOC significantly (Table 2). The interactions between soil C and N have been intensively studied as N could regulate plant photosynthesis, allocation, rhizosphere priming effect, and greenhouse gas emissions (*Gärdenäs et al., 2011*). For example, *Tian et al. (2019)* found that high N increased SOC mainly by decreasing $CO_2$ efflux. Organic carbon protected within microaggregates is isolated from microorganisms and hard to be decomposed. As a result, higher SA facilitates the storage of soil SOC. Unexpectedly, soil moisture, which is an important factor in regulating SOC decomposition (*Meyer, Welp & Amelung, 2018*), was insignificantly related to SOC in our study (Table 2). We measured SWC only one time, and it could not be representative of long-term conditions of soil moisture, which has a lasting impact on SOC decomposition. Higher soil pH was found in the middle-aged plantation than in the young and mature plantation (Table 1). In acidic soils, lower pH suppresses the growth of microorganisms and elevated pH could promote microbial biomass (*Silva-Sánchez, Soares & Rousk, 2019*). The higher pH and higher microbial biomass of middle-aged forests in this study are consistent. Although soil pH was not significantly related to SOC, pH can indirectly affect SOC through the regulation of microorganisms.

## CONCLUSIONS

Our study detected the variation in SOC of different stand ages, which has important implications for forest carbon sink function. Overall, middle-aged plantations had higher total SOC and organic carbon protected within microaggregates, which indicated that the SOC stability of middle-aged forests was stronger. Moreover, our study clarified that the physicochemical properties and microbial communities are the two main driving factors of SOC. The middle-aged plantation had significantly higher soil total PLFAs and all the functional groups of PLFAs than those of the young and mature plantations, indicating that the middle-aged plantation had better soil structure and nutrient status. In general, these findings jointly highlight the important role of stand age and soil biotic and abiotic factors in shaping the contents of SOC in subtropical *Cunninghamia lanceolata* plantations. These

results will greatly improve our understanding and prediction of soil carbon dynamics in forests with their development.

## ACKNOWLEDGEMENTS

We thank Shan Gao for his assistance in field and laboratory work. We also thank the academic editor and anonymous reviewers for their constructive comments, which helped in improving the manuscript.

### Funding

This work was financially supported by the National Natural Science Foundation of China (31700415), the Natural Science Foundation of Anhui Province (1808085QC60), and the Anhui Provincial Education Department (KJ2017A322). The funders had no role in study design, data collection and analysis, decision to publish, or preparation of the manuscript.

### Grant Disclosures

The following grant information was disclosed by the authors:
National Natural Science Foundation of China: 31700415.
Natural Science Foundation of Anhui Province: 1808085QC60.
Anhui Provincial Education Department: KJ2017A322.

### Competing Interests

The authors declare there are no competing interests.

### Author Contributions

- Ye Yuan conceived and designed the experiments, performed the experiments, prepared figures and/or tables, authored or reviewed drafts of the article, and approved the final draft.
- Juan Li performed the experiments, authored or reviewed drafts of the article, and approved the final draft.
- Liang Yao conceived and designed the experiments, analyzed the data, prepared figures and/or tables, and approved the final draft.

### Data Availability

   Raw data is available in the Supplementary Files.

### Supplemental Information

Supplemental information for this article can be found online at http://dx.doi.org/10.7717/peerj.13873#supplemental-information.

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
