# Peer review of "Soil microbial community and physicochemical properties together drive soil organic carbon in Cunninghamia lanceolata plantations of different stand ages"

_PeerJ, doi:10.7717/peerj.13873_

## Round 0.1 · original submission · Major Revisions

Dear Authors

Reviewers have now commented on your manuscript. They suggest a Major Revision. Kindly revise your paper as per the suggestions of the reviewers.

·

Basic reporting

Manuscript Number - (#74412) : In this paper, Author has demonstrated the “ Soil microbial community and physicochemical properties together drive soil organic carbon in Cunninghamia lanceolata plantations of different stand ages” The work is interesting, but following queries must be resolved before it publishing in the Journal.

 In Abstract- What is the means of microbial community composition? Some details of microbial community also mention.
 Kindly mention significant role of your work in abstract.
 In introduction –add some lines about Cunninghamia lanceolata plant.
 Authors should mention the description of the microbes
 In Methodology - Line 107-111 rewrite.
 In the section of “ Study site and soil sample collection” add references
 Add full form of PLFAs
 In result-Mention the name and species of microbial community
 The references used in this study are very old, mostly 2000-2003-2005,2009, 2011which
 should be updated
 After 2015 References add in MSS
 In figure 2,3 the statistically significant difference between all the treatments should be
Mentioned with different alphabets such as a, b, c, ab, cd……..
 Table 1, 2. mention the heading of first Colum

In last, this manuscript can be accepted after minor revision.

Experimental design

comment mention above.

Validity of the findings

Done

Additional comments

As per the title, please see your full MSS

·

Basic reporting

Please consider all PeerJ criteria to prepare the final style of manuscript structure.

Experimental design

no comment

Validity of the findings

in uploaded annotated PDF file

Additional comments

in uploaded annotated PDF file

---

## Round 0.2 · accepted · Accept

All the comments and suggestions of the reviewers have been addressed and the manuscript has been revised accodingly